# Hypertension Prevalence Based on Blood Pressure Measurements on Two vs. One Visits: A Community-Based Screening Programme and a Narrative Review

**DOI:** 10.3390/ijerph17249395

**Published:** 2020-12-15

**Authors:** Chloé Plumettaz, Bharathi Viswanathan, Pascal Bovet

**Affiliations:** 1Department of Health Services and Epidemiology, University Center for Primary Care and Public Health (Unisanté), 1010 Lausanne, Switzerland; ch.plumettaz@gmail.com; 2Ministry of Health, P.O. Box 52, Victoria, Seychelles; barathi.viswanathan@health.gov.sc

**Keywords:** blood pressure, hypertension, screening, surveillance, prevalence

## Abstract

We assessed the difference in the prevalence of hypertension in community surveys when blood pressure (BP) was measured on two vs. one visits and its impact on hypertension awareness, treatment and control proportions. A community-based BP screening programme was conducted in public places in the Seychelles (619 adults) and BP was rechecked a few days later among untreated participants with high BP (≥140/90 mmHg). A narrative review of the literature on this question was also conducted. Only 64% of untreated participants with high BP still had high BP at the second visit. The prevalence of hypertension in the whole sample decreased by 13% (from 33.8% to 29.5%) when BP was measured on two vs. one visits. These results concurred with our findings in our narrative review based on 10 surveys. In conclusion, the prevalence of hypertension can be markedly overestimated in community surveys when BP is measured on two vs. one visits. The overestimation could be addressed by measuring BP on a second visit among untreated individuals with high BP or, possibly, by taking more readings at the first visit. These findings have relevance for clinical practice, policy and surveillance.

## 1. Introduction

Hypertension is the leading cause of the global disease burden, accounting for 10.4 million deaths in 2017 [1] and contributing to 22.3% of the population attributable burden of cardiovascular disease [2]. The prevalence of high blood pressure (BP) ranges between 30–50% in adults in many populations worldwide [3]. The World Health Organization (WHO) Global Action Plan for the Prevention and Control of Noncommunicable Diseases (NCDs) includes the target to reduce by 25% the prevalence of raised BP between 2010 and 2025 and summarizes cost-effective interventions to reduce its burden at both population and individual levels [4].

BP tends to decrease over repeated readings within and across visits [5], owing to habituation to BP measurements and regression to the mean. For example, a review of clinical studies found that 5–65% of persons who had high BP on one visit did not have hypertension when BP was measured with ambulatory blood pressure monitoring (27 studies) and 4–42% did not have hypertension when BP readings were repeated at subsequent office visits (4 studies) [6]. This stresses that hypertension diagnosis should be based on several BP readings taken on several occasions (at least two), as recommended by different international guidelines [7,8,9].

Similarly, the prevalence of hypertension in community surveys (as opposed to the prevalence of “high blood pressure”) can be assessed more reliably if BP is measured on several visits, as already pointed out in 1969 [10]. A Canadian cross-sectional community survey showed that, while the number of BP readings made at a single visit can influence prevalence estimates (e.g., when the prevalence is based on the average of three readings or the last two of three readings), the prevalence decreased most (by 12% and 17%) when based on BP readings measured again at two or three subsequent visits [11]. Similarly, the prevalence of hypertension decreased by as much as 48% when BP was measured on a second visit taking place at least 2 years after the first visit in a recent Chinese study, independently of hypertension treatment [12].

However, due to logistic difficulties and cost, BP is measured on one single visit in most community surveys, e.g., WHO STEPS [13] or the May Measurement Month (MMM) screening programme [14]. Two reviews of population-based surveys of hypertension showed that only 4 out of 44 surveys [15] and 1 out of 68 [16] had assessed BP on at least 2 visits, as opposed to BP measured on 1 visit only.

In this study, we assessed the difference in estimates of the prevalence of hypertension when BP was measured on two visits vs. one visit in a community-based BP screening programme in the Seychelles and the impact on hypertension awareness, treatment and control proportions in the underlying population, an issue that has only rarely been examined [11]. We also conducted a narrative review of all published papers on community surveys that had assessed the prevalence of hypertension based on BP measurements on ≥two vs. one visits.

## 2. Materials and Methods

We analysed BP data of a community-based screening programme led by the Ministry of Health in the Seychelles performed around World Hypertension Day every year. We included data from the participants in 2018 and 2019. The screening programme took place in several public places in the main island (Mahe, 90% of the total population of the Seychelles) between 9 a.m. and 2 p.m. The methods used in the screening programme to measure BP were consistent with hypertension guidelines [7,8,9] and with the May Measurement Month protocol [17,18].

The blood pressure screening was offered free to any volunteer adults aged 18 years or more. Each participant was asked for their consent to use their anonymous data for research. A short-structured questionnaire about socioeconomic indicators and lifestyle variables was administered to the participants. Weight was measured using an electronic scale (seca 703^®^, seca GmbH & Co. KG, Hamburg, Germany), height was measured using a fixed stadiometer (seca 220^®^, seca GmbH & Co. KG, Hamburg, Germany), and random blood glucose was measured on capillary blood using glucometer (Contour Plus^®^, Bayer Healthcare AG, Leverkusen, Germany). Overweight and obesity were defined as a body mass index (BMI) of 25–29.9 kg/m^2^ and ≥30 kg/m^2^, respectively. Raised blood sugar was defined as random blood glucose ≥7.8mmol/L, according to the American Diabetic Association guidelines [19].

BP was measured on individuals comfortably seated in a chair with their left arm laying on a table at the heart level, using a validated electronic device (Omron M6 Comfort^®^, Omron Healthcare Co., Ltd., Kyoto, Japan) [20]. After a rest of ≥3 min, BP was measured 3 times at intervals of ≥1 min. The screening programme was led and approved by the Ministry of Health. Participants were informed about the screening procedures and asked for their written consent to use their data in aggregate form for evaluative research.

Individuals with systolic BP ≥140 mmHg and/or diastolic BP ≥90 mmHg (based on the average of 3 readings) and not taking any treatment for hypertension were invited to attend a second visit a few days later. Participants could indicate their real name or a fake name on their appointment card in order to keep their data anonymous while ensuring a link with the data from the first visit. The second visit took place in a health promotion unit in the ministry of health where health workers wore civil garments. On the second visit, BP was checked by the same nurse who had seen the participant at the first visit, using the same measurement protocol and BP measuring device. No treatment for hypertension was prescribed between the visits. The distribution of mean BP and the prevalence of raised BP were tabulated, and 95% confidence intervals were calculated.

The study was approved and conducted by the Ministry of Health of the Seychelles. BP readings at Visit 1 (but not at Visit 2) have been included in previous papers from the May Measurement Month global study in 2018 and 2019 [17,18].

We also performed a narrative review of all published papers or reviews examining the difference in the prevalence of hypertension in population-based surveys when BP was measured on ≥2 vs. 1 visits among adults. We searched all full-text papers published in English and in French up to January 2020 available in PubMed, Cochrane Library and Google Scholar using the following descriptors: “Hypertension, prevalence, repeated visits, two visits, repeated measurements, serial surveys.” We also identified further pertinent publications from the lists of references. We included all papers that had the following criteria: A population-based sample (community surveys); the prevalence of hypertension was estimated based on BP readings taken on the first and second visits; BP measurement was based on the average of ≥2 readings at each visit; the same measurement methods were used at each visit. When a study did not report explicitly the relative difference in the prevalence of hypertension when estimates were based on reading on ≥2 vs. 1 visits, the relative difference was estimated based on the prevalence values indicated in the study (in the tables or figures). Studies that provided only data on systolic BP [21,22], diastolic BP [23,24,25,26], using a single BP measurement on a particular visit, giving the estimation of hypertension prevalence based on the sole reading on the final visit [27], having an interval between visits exceeding 2 months [12] or were not community-based surveys [28,29] were excluded. We also excluded the papers for which only the abstracts were available in the 3 searched library databases [30,31,32,33]. Several studies included in the previously mentioned 3 reviews [6,15,16] were excluded because they did not meet all of our inclusion criteria. In one instance, the results of a potentially suitable study were published in both French [34] and English [35] in different journals and we only included the publication in English in our review. We identified 51 studies that seemed to have assessed hypertension prevalence in community survey based on ≥2 vs. 1 visits [6,10,11,12,15,16,21,22,23,24,25,26,27,28,29,30,31,32,33,34,35,36,37,38,39,40,41,42,43,44,45,46,47,48,49,50,51,52,53,54,55,56,57,58,59,60,61,62,63,64,65], and 10 met our inclusion criteria (Table 6).

## 3. Results

A total of 620 individuals participated in our community-based screening programme (age range: 18 to 86 years). One participant was excluded because of missing data. Seventy-three participants had systolic BP ≥140 mmHg and/or diastolic BP ≥90 mmHg and were not treated for hypertension. Among them, 50 (69%) had their BP measured on a second visit. The mean interval between the first and second visits was 2.6 days (range: 1 to 12 days; 95% CI: 2.1–3.1).

Table 1 shows selected characteristics of the participants at the first visit according to age and sex. There were 241 men (38.9%) and 378 women (61.1%), and 320 were aged <45 years (51.7%) and 299 were ≥45 years (48.3%). Almost all participants (96.9%) had their BP measured at least once in their life and 86.1% had it checked within the past 12 months, with the highest proportion among women over 45 years old (92.0%).

One-third of the participants (33.8%) had high BP (BP ≥140/90 mmHg or treatment) at the first visit, with differences according to age (17.1% at age <45 years and 48.2% at age ≥45 among men, and 15.2% and 55.6% among women of the same age categories). More than half of participants with high BP were taking an antihypertensive treatment (men 52.6%; women 72.2%). In the whole sample, 22% of the participants were taking a BP-lowering treatment, particularly those aged ≥45 years (37.5%) and women aged ≥45 years (43.3%). The prevalence of smoking was higher in men than women (25.7% vs. 3.4%). Three-quarters of men and women were overweight or obese. More women than men were obese (40.2% vs. 27.4%). Around 8% had diabetes (with a higher proportion in the older than younger age groups).

Table 2 shows that mean systolic BP and mean diastolic BP decreased between the first, second and third readings within each visit, with most of the decrease occurring between the first and second BP readings. The difference between the first and third readings within each visit, for both systolic and diastolic BP, was approximately 3 mmHg in each BP subgroup (i.e., participants with BP <140/90 mmHg, participants treated, participants with BP ≥140/90 mmHg but untreated, and participants seen on the second visit). The difference reached statistical significance in the BP categories with the largest numbers of participants.

Table 3 shows the mean systolic/diastolic BP (based on the average of the 3 readings at each visit) among the 50 participants with BP ≥140/90 mmHg and not treated who attended the second visit. BP significantly decreased between the first and second visits by 5.6/3.3 mmHg for systolic/diastolic BP, with a relative drop of respectively 3.5%/3.3% for systolic/diastolic BP, respectively. This decrease was of a similar magnitude in both sexes and age groups.

Table 4 shows that 67.9% of men and 59.1% of women with BP ≥140/90 mmHg and untreated at the first visit also had elevated BP (i.e., BP ≥140/90 mmHg) at the second visit. Inversely, around one-third of untreated participants with elevated BP at the first visit no longer had elevated BP at the second visit. In relative terms, the prevalence of hypertension (BP ≥140/90 mmHg or treatment) decreased by 12.6% (men 15.2%, women 11.4%) based on BP readings on two vs. one visits. This estimate assumed the same decrease in BP between the 2 visits among the 28 participants with BP ≥140/90 mmHg and no treatment who attended the second visit as in the 8 participants with hypertension who did not attend the second visit.

Table 5 shows the proportions of persons aware, treated and controlled among all persons with hypertension (BP ≥140/90 mmHg or treatment) based on two vs. one visits. Since the denominator of these proportions (i.e., the total number of persons with defined hypertension) was lower when hypertension was assessed based on BP readings on two vs. one visits, the proportions of persons aware, treated and controlled, in the entire sample, was higher by 14.4% (men 18.0%; women 12.8%) when the diagnosis of hypertension was based on two vs. one visits.

Table 6 shows the 10 studies included in our narrative review of the difference in the prevalence of hypertension based on ≥2 vs. 1 visits [11,35,36,37,38,39,40,41,42,43]. The included studies are displayed according to the number of visits used to define hypertension. Mean BP was lower by 1.5–7.0 mmHg (systolic BP) and 0.7–9.0 mmHg (diastolic BP) when comparing mean BP based on two vs. one visits. The decrease in mean BP could be larger in those studies in which BP was measured on more than two visits. The relative difference in the prevalence of hypertension when BP was measured on two vs. one visits ranged between 12% and 39%, and the difference between the last and first visit could be larger when BP was measured on more than two visits (up to 43%).

## 4. Discussion

Based on our community-based screening programme, the prevalence of hypertension was 13% lower when BP was measured on two vs. one visits. This estimate seems conservative compared to the results from our narrative review of 10 community surveys, where the prevalence of hypertension decreased, in relative terms, by 12% to 43% between the last vs. first visits [11,35,36,37,38,39,40,41,42,43]. Despite the difficulty of getting an accurate overall estimate of the impact measuring BP readings on one vs. several visits on the prevalence of hypertension in populations due to the discrepancy between the methods used in the studies (number of BP readings, intervals between visits, etc.), our narrative review included the largest number of eligible such studies and provides the most solid evidence on this question so far. We also found that the proportions of hypertension awareness, treatment and control rates in the sample of participants to the screening programme were substantially larger (by approximately 14% in our study) when hypertension (i.e., the denominator of these proportions) was based on two vs. one visits. Our findings re-emphasize that the prevalence of hypertension in population-based surveys can be substantially overestimated when BP is measured on ≥two vs. one visits, and awareness, treatment and control rates (in the underlying populations) can consequently be substantially underestimated when based on BP readings taken on ≥two vs. one visits.

As expected, BP decreased over three consecutive readings at each visit (with a difference of approximately 3 mmHg between the first and third readings in our study), irrespective of initial BP level and treatment status, consistent with findings in several other studies [5,38,39,42,47,66]. In addition, mean BP (based on triplicate readings at each visit) was lower at the second vs. first visits by approximately 5 mmHg, which is consistent with findings in our review (Table 6) [38,39,42]. The BP decrease over repeated readings at a same visit and between subsequent visits is largely attributed to habituation to BP measurement [42]. Of note, the marked systematic decrease in mean BP over subsequent visits does not necessarily apply to other cardiovascular risk factors such as body weight, blood cholesterol or blood glucose, as these latter risk factors do not necessarily systematically decrease over consecutive visits. This underlies that hypertension guidelines stress that the diagnosis of hypertension should be based on BP readings taken over several visits.

It is likely that the decrease in the prevalence of hypertension based on two vs. one visits may differ between populations. First, this decrease is likely to differ according to the absolute mean BP level in the underlying population. Indeed, the decrease in the prevalence of hypertension (or the difference in mean BP level) based on two vs. one visits is likely to be proportionally larger if the average BP in the underlying population is close to the cut-off value for elevated BP, as opposed to populations where the mean BP would be markedly lower than 140/90 mmHg, because more persons in the population would have their BP levels near the cut-off for hypertension in the former vs. latter instances [47]. In our study, the prevalence of hypertension (defined in this study as BP ≥ 140/90 mmHg) and mean BP levels in the underlying population were quite high compared to some other countries (e.g., North America) but were similar with several countries in Sub-Saharan Africa [17,18].

The decrease in the prevalence of hypertension based on BP measurements made on two vs. one visits might also be larger if the proportions of untreated persons with elevated BP is high vs. low, as a larger proportion of individuals in the underlying populations could possibly normalize their BP over consecutive visits in the former situation. The proportion of treated persons was fairly high in the Seychelles, at 22% of the whole population in our study, which is similar with a recent population-based survey of cardiovascular risk factors in the country [67]. This implies that the difference in the prevalence of hypertension in surveys based on two vs. one visits could be larger in countries where only few people with hypertension are treated for hypertension. Further studies should assess the prevalence of hypertension based on two vs. one visits in countries with different mean BP levels and different proportions of hypertensive persons treated.

Another issue, not addressed in our study, is to clarify whether all persons treated for hypertension actually had hypertension at the first place. It may happen that some persons are (inadequately) given hypertension medication(s) on the basis of elevated BP taken on a single visit, whereas a certain proportion of them would not have had sustained hypertension if BP had been measured on ≥two visits (i.e., false-positive hypertension cases). This situation would lead to overestimation of the hypertension prevalence in epidemiological surveys. Further studies should assess the proportions of false-positive cases in population-based surveys or in cohort studies in different populations using treatment de-intensification in selected treated persons. However, this would require several follow-up visits over several months. For example, a review of studies of hypertension treatment de-intensification found that approximately one-quarter of patients on antihypertensive therapy were unnecessarily treated and assessed with no relapse of hypertension after ≥two years of treatment discontinuation [68].

BP can further decrease when BP is based on ≥two visits vs. only one or two visits or, quite similarly, if ambulatory BP monitoring or home BP monitoring (with multiple readings over an extended period of time) is performed among patients with elevated BP and no treatment. For example, the prevalence of hypertension decreased further when BP was measured on three vs. two visits in two population-based surveys in Tanzania [42] and Nigeria [39].

Another issue is the impact of a white coat effect on the prevalence of hypertension in epidemiological studies, whereby BP is artificially increased when BP is measured by medical personnel [41,69]. This factor can increase the overestimation of the prevalence of hypertension in epidemiological studies, but this overestimation could possibly be partially compensated by the effect of masked hypertension on the prevalence of hypertension [69]. In this study, BP was measured in secluded places (first visit) and in an administrative building (second visit) by nurses wearing normal clothes (i.e., not a medical uniform), and one could expect that the white coat effect was minimized [70].

The decreased number of hypertensive persons based on two vs. one visits (this number being used in the denominator) resulted in increased proportions of persons aware, treated and controlled for hypertension in the underlying population when the prevalence of hypertension was based on BP readings taken on two vs. one visits (a relative increase of 18% in our study). We are not aware of other studies that have explicitly quantified this difference [11]. The issue is important as it implies that the usually low proportions of awareness, treatment and control found in many epidemiological surveys that have assessed BP on a single visit might not be as low as they appear if the prevalence of hypertension was assessed based on BP readings taken on several visits. Further studies on awareness, treatment and control rates should attempt to include a more valid denominator (i.e., the number of persons with hypertension based on two or more visits) or, at least, such studies should attempt to explicitly discuss and quantify the possible underestimation in these awareness, treatment and control proportions when hypertension is assessed based on BP measured on ≥two vs. one visits.

This study examined the validity of the diagnosis of hypertension when BP was measured on two vs. one visits. However, correctly assigning hypertension diagnoses to persons who have BP permanently ≥140/90 mmHg (or another BP cut-off) does not imply that it is inadequate to prescribe BP-lowering treatment to selected persons with lower BP levels but high cardiovascular risk (based on other elevated risk factors) in order to reduce their total cardiovascular risk (e.g., total risk approach, “polypill” interventions, etc.) [71]. Nonetheless, a valid assessment of the prevalence of hypertension (based on BP readings measured on several visits) in epidemiological surveys is important for both surveillance and policy purposes.

Our study has several strengths. First, participants from our community-based screening programme included adults from the general population. Although participants were not randomly selected from the general population, the population-based nature of the sample is supported by the fact that the prevalence of hypertension in the screening programme was similar to that found in a national survey of BP and other risk factors in the Seychelles [67]. Second, untreated participants with high BP in the screening programme had their BP rechecked on a second visit. With regard to our review, no previous paper had made a systematic review of the prevalence of hypertension in community surveys based on BP readings on two vs. one visits. Our study and narrative review also have limitations. First, the sample of our case study was fairly small. Yet, the sample is likely sufficiently large to indicate the approximate magnitude of the estimates under study, and our review of the literature concurred with our results. Second, we did not assess the validity of the hypertension diagnosis in persons in the screening programme treated for hypertension (some of whom might not have been hypertensive in the first place, up to one-quarter in some studies [68]). Third, we did not assess the impact of measuring BP on more than two visits in our screening programme. Fourth, we did not assess the impact of using different combinations of BP readings at each visit (e.g., last two of three readings, two readings, more readings) for the sake of simplicity and because these factors may have a marginal impact as compared to differences in BP readings taken on several visits [11]. With regard to our narrative review, we did not attempt to meta-analyse an overall estimate of the overestimation of hypertension based on two vs. one visits, as the included epidemiological studies have used largely different methods (definition of hypertension, number of BP readings per visit, participants eligible for repeat visits, mean BP level in the underlying population, etc.).

## 5. Conclusions

In conclusion, our community screening programme and our narrative review re-emphasize that the prevalence of hypertension in the underlying population is markedly lower when based on BP readings measured on two vs. one visits. From an individual-based perspective, it is sometimes argued that defining the diagnosis of hypertension based on a single visit can still be useful to avoid suspected hypertension cases not coming back for confirmatory measurements and/or treatment on subsequent days. In Tanzania, less than one-third of the newly identified hypertensive persons from a large epidemiological survey attended at least one further medical visit during the next 12 months despite being advised to do so, and in the Seychelles, only 20% of newly discovered hypertension cases from a national population-based survey regularly took their prescribed medication after 12 months [44,45]. Whereas a low uptake of adequate BP management for a condition that generally shows no symptoms is unfortunate (and needs to be addressed), a valid diagnosis of hypertension is important to avoid unnecessary worries and treatment among false-positive cases (clinical perspective), as well as for adequate surveillance of the prevalence of hypertension and related health policy (epidemiological perspective). The validity of hypertension prevalence estimates in epidemiological studies could be improved by measuring BP on a second visit (or, possibly, by taking additional BP readings at a first visit) for all persons or for a random sample of persons with elevated BP and no treatment at the first visit [5,47] or, at least, by attempting to quantify the possible overestimation of the prevalence of hypertension in surveys in which BP was measured on one single visit based on estimates of the relative reduction of the prevalence of hypertension from studies that have measured BP on more than one visit [41,72].

## Figures and Tables

**Table 1 ijerph-17-09395-t001:** Characteristics of the participants to the community-based screening programme according to sex and age.

	Men			Women			All		
	<45 Year	≥45 Year	Total	<45 Year	≥45 Year	Total	<45 Year	≥45 Year	Total
Participants (*n*)	129	112	241	191	187	378	320	299	619
Ever had BP checked (%)	92.2	98.2	95.0	98.4	97.9	98.1	95.9	98.0	96.9
Had a BP check in past 12 months (%)	76.7	86.6	81.3	86.4	92.0	89.2	82.5	90.0	86.1
BP ≥140/90 mmHg or treated (%)	17.1	48.2	31.5	15.2	55.6	35.2	15.9	52.8	33.8
From which % treated	40.9	57.4	52.6	51.7	77.9	72.2	47.1	70.9	65.1
Treated for hypertension (%)	7.0	27.7	16.6	7.9	43.3	25.4	7.5	37.5	22.0
Smoking (%)	31.0	19.6	25.7	4.2	2.7	3.4	15.0	9.0	12.1
Body mass index ≥25 kg/m^2^ (%)	73.6	76.8	75.1	67.0	83.4	75.1	69.7	80.9	75.1
Body mass index ≥30 kg/m^2^ (%)	24.0	31.3	27.4	34.6	46.0	40.2	30.3	40.5	35.2
Aware of having diabetes (%)	0.8	8.0	4.1	1.6	9.1	5.3	1.3	8.7	4.8
Random blood glucose ≥7.8 mmol/L (%)	6.2	11.6	8.7	4.2	10.7	7.4	5.0	11.0	7.9

Blood pressure (BP) estimates are based on the average of three readings.

**Table 2 ijerph-17-09395-t002:** Mean systolic and diastolic blood pressure on three successive blood pressure readings according to hypertension status and different visits in the community-based screening programme.

			Reading 1		Reading 2		Reading 3		Relative diff. (%)	
		*n*	Mean	95%CI	Mean	95%CI	Mean	95%CI	R3 vs. R1	95%CI
Visit 1 (*n* = 619)										
BP <140/90, no Rx	Sys	410	115.5	114, 117	112.8	112, 114	112.0	111, 113	−3.0	−5, −1
	Dia		76.6	76, 77	75.3	74, 76	74.1	73, 75	−3.3	−5, −1
Treated	Sys	136	133.1	130, 136	130.1	127, 133	128.5	126, 131	−3.5	−7, −0.4
	Dia		86.4	84, 88	84.4	82, 86	83.7	82, 86	−3.1	−6, −0.3
BP ≥140/90, no Rx	Sys	73	146.0	142, 150	142.4	139, 146	141.9	139, 145	−2.8	−7, 1
Visit 2 * (*n* = 50)	Dia		96.3	94, 98	94.2	92, 96	92.8	91, 94	−3.6	−8, 1
BP ≥140/90 at V1, no Rx	Sys	50	144.8	141, 148	142.0	138, 146	140.1	137, 144	−3.2	−8, 2
	Dia		94.5	92, 97	92.0	90, 94	90.0	87, 93	−4.8	−11, 1

BP: Blood pressure (mmHg); Sys: Systolic BP; Dia: Diastolic BP; V1: Visit 1. Rx: Treatment for HTN: Hypertension. * Visit 2 was attended only by participants with BP ≥140/90 at Visit 1 and no treatment.

**Table 3 ijerph-17-09395-t003:** Mean systolic/diastolic blood pressure on the first and second visits among participants to the community-based screening programme who were untreated and had elevated blood pressure at the first visit.

			Visit 1		Visit 2 *		Absolute dif. V2 vs. V1		Rel. dif.
	*n*		BP	95%CI	BP	95%CI		95%CI	%
Men	28	Sys	147.8	143, 153	142.8	139, 147	−5.0	−11, 1	−3.0
		Dia	95.5	92, 99	93.2	90, 97	−2.3	−7, 2	−2.8
Women	22	Sys	148.1	141, 155	141.6	136, 148	−6.5	−16, 3	−4.1
		Dia	96.6	93, 99	90.9	88, 93	−4.7	−9, −1	−4.6
<45 years	14	Sys	143.7	136, 152	136.1	131, 141	−7.6	−17, 2	−4.9
		Dia	96.1	92, 101	91.5	88, 95	−4.6	−10, 1	−4.6
≥45 years	36	Sys	149.5	145, 154	144.7	140, 149	−4.8	−11, 1	−3.0
		Dia	95.3	93, 98	92.5	90, 95	−2.8	−7, 1	−2.8
Total	50	Sys	147.9	144, 152	142.3	139, 16	−5.6	−11, −0.3	−3.5
		Dia	95.5	93, 98	92.2	90, 94	−3.3	−7, −0.2	−3.3

BP: Blood pressure; Sys: Systolic BP; Dia: Diastolic BP; Rel. dif: Relative difference (%). *** Visit 2 was attended only by participants with BP ≥140/90 at Visit 1 and no treatment.

**Table 4 ijerph-17-09395-t004:** Prevalence of hypertension in the community-based screening programme based on blood pressure measured on two vs. one visits.

	Men	Women	Total
Number with BP <140/90, untreated, Visit 1	165	245	410
Number treated, Visit 1	40	96	136
Number with BP ≥140/90, untreated, Visit 1	36	37	73
Number with BP ≥140/90, untreated, Visit 1 & had BP measured at Visit 2	28	22	50
Number with BP ≥140/90, untreated, Visit 1 & had BP ≥140/90 at Visit 2	19	13	32
* Expected number with BP ≥140 /90, untreated, V1 & BP ≥140/90 at V2	24.4	21.9	46.7
***Proportion of participants untreated and with BP*** **≥*140/90 at Visit 1 who also had BP* ≥*140/90 at Visit 2***	***67.9***	***59.1***	***64.0***
Prevalence (%) of hypertension in the whole sample based on BP readings at Visit 1	31.5	35.2	33.8
Prevalence (%) of hypertension in the whole sample based on BP readings at Visit 1 & Visit 2	26.7	31.2	29.5
***Relative decrease in the prevalence of hypertension in the whole sample based on BP measured on two vs. one visits (%)***	***−15.2***	***−11.4***	***−12.6***

BP: Blood pressure; hypertension: BP ≥140/90 mmHg (average of 3 readings) or treatment. Visit 2 (V2) was attended only by untreated participants with BP ≥140/90 at Visit 1 (V1). * The differences in BP between V2 vs. V1 assume a same relative BP change in eligible participants who attended V2 as in those who did not attend V2.

**Table 5 ijerph-17-09395-t005:** Proportions of participants in the community-based screening programme who were aware, treated and controlled for hypertension based on the total number of persons with hypertension was based on blood pressure measured on two vs. one visits.

	Men	Women	All
Number of participants	241	378	619
Number with hypertension based on BP measured at V1	76	133	209
Number with hypertension based on BP measured at V1 & V2	64	118	183
Number aware of having HBP	50	112	162
% aware from all participants with HBP, based on V1 (A)	65.8	84.2	77.5
% aware from all participants with HBP, based on V1 & V2 (B)	77.6	95.0	88.7
***Relative difference (B vs. A) (%)***	***18.0***	***12.8***	***14.4***
Number treated for hypertension	40	96	136
% treated from all participants with HBP, based on V1 (C)	52.6	72.2	65.1
% treated from all participants with HBP, based on V1 & V2 (D)	62.1	81.5	74.4
***Relative difference (D vs. C) (%)***	***18.0***	***12.8***	***14.4***
Number with BP controlled (<140/90 mmHg)	21	63	84
% controlled from all participants with HBP, based on V1 (E)	27.6	47.4	40.2
% controlled from all participants with HBP, based on V1&V2 (F)	32.6	53.5	46.0
***Relative difference (F vs. E) (%)***	***18.0***	***12.8***	***14.4***

V1: Visit 1; V2: Visit 2; BP: Blood pressure; HBP: High blood pressure. Hypertension: BP ≥140/90 mmHg (based on average of 3 readings at each visit) or treatment. BP was measured at Visit 2 among untreated participants with BP ≥140/90 at Visit 1.

**Table 6 ijerph-17-09395-t006:** Narrative review: Characteristics and main results of the community surveys that had assessed the difference in the prevalence of hypertension based on blood pressure measured on ≥two vs. one visits.

First Author Nb Visits [ref]	Place of Study, Year of Data Collection	Sample Size	Nb Readings at Each Visit	Hypertension Definition	Criterion for Subsequent Visit	Mean BP		Rel. dif. HTN Prevalence
						First Visit	Last Visit	
2 visits								
Inamo J [35]	Martinique/Guadeloupe/Guyana, 2001–2002	6113	1—2—3	≥140/90 or Rx	≥140/90, no Rx	NA	NA	−19%
Lang T [36]	France, 1997–1998	29,626	1—2—3	≥140/90 or Rx	≥140/90, no Rx	NA	NA	* −39%
Atallah A [37]	Guadeloupe, 2007 CONSANT survey	1005	1—2—3	≥140/90 or Rx	≥140/90, no Rx	NR	NR	* −25%
	Guadeloupe, 2001–2003 PHAPGG survey	20,420	1—2—3	≥140/90 or Rx	≥140/90, no Rx	NR	NR	* −32%
Modesti PA [38]	Yemen, 2009	10,242	2—3	≥140/90 or Rx	All	123.0/76.9	121.5/76.2	−33%
Markovic N [39]	Nigeria, 1992	804	2—3	≥140/90 or Rx	All	* 120.5/75.4	* 116.9/85.0	NR
Pierce L [40]	Haiti, 2012	175	1—2	≥140/90 or Rx	≥140/90, no Rx	NR	NR	* −20%
Figueiredo D [41]	Portugal, 2001–2003	739	1—2	≥140/90 or Rx	All	NR	NR	−13%
				≥140/90, no Rx	All	NR	NR	−31%
Bovet P [42]	Tanzania, 1998–1999	9254	2—3	≥160/95	All	* 147.3/89.3	* 140.2/85.8	−29%
Birkett NJ [11]	Canada, 1985–1986	2016	1—2—3/2—3	≥160/90 or Rx	≥160/90 or Rx	NA	NA	* −12%
**3 visits**								
Lai M [43]	China, 2017	1185	1—2—3	≥140/90 or Rx	All	NR	NR	* −12%
Markovic N [39]	Nigeria, 1992	804	2—3	≥140/90 or Rx	All	* 120.5/75.3	* 115/73.3	* −14%
Birkett NJ [11]	Canada, 1985–1986	2016	1—2—3/2—3	≥160/90 or Rx	≥160/90 or Rx	NA	NA	* −17%
**4 visits**								
Bovet P [42]	Tanzania, 1998–1999	9254	2–3	≥160/95	All	* 147.3/89.2	* 136/83.9	−43%

NA: Not applicable; NR: Not reported; BP: Blood pressure; HTN: Hypertension; Rx: Treatment; Rel. dif.: Relative difference; nb: Number; ref: Reference. * Difference in the prevalence of hypertension based on ≥two vs. one visits based on prevalence figures displayed in the article.

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
