# Peer review of "Hypertension Prevalence Based on Blood Pressure Measurements on Two vs. One Visits: A Community-Based Screening Programme and a Narrative Review"

_ijerph, 2020, doi:10.3390/ijerph17249395_

Round 1

Reviewer 1 Report

This study has explored the difference in the prevalence of hypertension in community surveys when blood pressure (BP) is measured on 2 vs 1 visits. The authors also conducted a narrative review on this topic. Some comments are listed below:

Introduction: the innovations of the study should be talked more.

Methods: the ethnical statement and informed consent should be stated.

Methods: Were there any inclusion and exclusion criteria for the enrollment of the subjects?

Methods: some statistical analysis methods (ICC, t-test, χ2) can be used to estimate the accordance and differences of the BP between different readings or visits.

Methods: Why was the data presented by age group of <45 years and ≥ 45 years?

Discussion: As the narrative review is also a part of the present study, a comprehensive discussion of the published studies is suggested, not just listed the information of those studies in Table 6.

Discussion: the conclusion can be more concise.

References: the order of the references is difficult to follow and some references are not cited in the main text.

Author Response

Point 1. Introduction: the innovations of the study should be talked more.

Response 1: We thank the reviewer for the comment. The primary question addressed by the study (“what is the proportion of persons with HBP on a visit that still have HBP on a second visit”) is not new but is of extremely great importance (i.e. distinction between HBP vs hypertension). However, as we mention in the introduction, few studies have assessed this question in the epidemiologic settings (i.e. in a randomly selected population, as opposed to clinical setting), as we demonstrate in our narrative review. This question is central to the interpretation of the “prevalence of hypertension” in population. As we mention, most epidemiological BP surveys (in random samples of the population) reported only the prevalence of HBP based on BP readings on only 1 visit. The question, then, is: to what extent the prevalence of HBP (based on readings on 1 visit) predicts the prevalence of hypertension (based on readings on 2 visits). Our study brings further evidence that the difference is important (i.e. prevalence of hypertension is substantially lower than the prevalence of HBP) and the implication is that studies that assess the prevalence of HBP (based on reading on 1 visit) should discuss this issue explicitly (which very few studies do). The secondary goal of the study was to quantify the rates of BP awareness, treatment and control when the denominator is the prevalence of HBP vs the prevalence of hypertension. We are not aware of other studies that have assessed this important question.

Point 2. Methods: the ethical statement and informed consent should be stated.

Response 2: Before the screening, each participant was asked for their consent to participate in the screening and to use their anonymous data for research purposes (sentence added on page 2 line 67).

The study was approved by the Ministry of Health of the Seychelles and conducted by the staff of the Ministry of Health. (sentence added on page 2 line 89).

Point 3. Methods: Were there any inclusion and exclusion criteria for the enrollment of the subjects?

Response 3: The screening of blood pressure was offered for free to any volunteer adult aged 18 years or more (sentence added on page 2 line 66). The only exclusion criteria in this study was age ≤ 18 years.

Point 4. Methods: some statistical analysis methods (ICC, t-test, χ2) can be used to estimate the accordance and differences of the BP between different readings or visits.

Response 4: The stated primary goal of this study was to compare the prevalence of HBP (based on BP measurements on 1 visit) vs the prevalence of hypertension (based on BP readings on 2visits), or, similarly, compare the prevalence of awareness, treatment and control of BP when the denominator includes person with HBP (based on 1 visit) vs persons with hypertension (based on 2 visits). Hence, the most useful statistic for this purpose is the 95% CI, as used in the study (pages 3 and 4) and tables 3 (page 7) and 4 (page 8).

Point 5. Methods: Why was the data presented by age group of <45 years and ≥ 45 years?

Response 5: We wanted to stratify results by age and sex. The relatively small sample prevented to use more age categories. These 2 categories can fairly well represent “young” and “older” adults.

Point 6. Discussion: As the narrative review is also a part of the present study, a comprehensive discussion of the published studies is suggested, not just listed the information of those studies in Table 6.

Response 6: Thank you for the remark. We agree that more information might be provided on characteristics and results of the referred studies but this would considerably extend the length of the paper. However, consistent with the goal of our case study in Seychelles, the main purpose of the narrative review was to compare the prevalence of HBP (based on BP reading on 1 visit) vs the prevalence of hypertension (based on 3 or more visits) in other studies (and other populations) in the literature. The table of results of this narrative review is as concise as possible but it does provide the desired information. Also, as we mention in the paper, we did a narrative review (i.e. not a meta-analysis) as studies differed considerably in terms of methods. Our narrative review is not aimed at providing a summary estimate of the difference in the prevalence of HBP vs prevalence of hypertension, but merely an order of magnitude of this difference. We found that this order of magnitude was large, further supporting our recommendation that epidemiologic studies that assess the prevalence of “hypertension” based on BP based on only 1 visit should discuss this issue of overestimation of “hypertension” when BP is measured based on 1 visit vs 2 visits.

Point 7. Discussion: the conclusion can be more concise.

Response 7: Thank you for the remark. We believe that the conclusion well summarizes the various main take-home messages and recommendations implied by our study and the narrative review. But we are prepared to shorten the conclusion if the reviewer or the editor requires it.

Point 8. References: the order of the references is difficult to follow and some references are not cited in the main text.

Response 8Thank you for the remark. We have checked all the references and their location in the main text/tables so that they conform with the Journal’s requirements.

Reviewer 2 Report

In their manuscript, Plumettaz et al. performed a community survey of the prevalence of hypertension in Seychelles where blood pressure (BP) was measured on 2 separate occasions.  During each visit, BP was measured 3 times and the average of the 3 measurements was taken as the BP for that visit. They found that blood pressure decreased by an average of 3 mmHg between the first and third reading at each visit. In addition, in untreated individuals whose BP was >140/90 at the first visit, the prevalence of hypertension decreased by 13% at the second visit.

-International guidelines on the diagnosis of hypertension state that BP has to be elevated at a minimum of 2 visits (2 different occasions), therefore the diagnosis of hypertension should not be made on the basis of a single BP measurement to begin with.

-In table 3, when divided into groups, the 95% confidence interval for the absolute difference in BP between visit 1 and visit 2 crosses 0, suggesting no statistical significance.  In general, statistical significance (or lack thereof) needs to be better indicated throughout the manuscript.

-For table 4, the assumption the authors make about BP reduction in untreated individuals with elevated BP at visit 1 who did not have a second visit is unclear and I don’t think it’s valid (page 3 line 145).  Rather, the data should only be presented for those 50 individuals who had 2 visits.

Author Response

Point 1. International guidelines on the diagnosis of hypertension state that BP has to be elevated at a minimum of 2 visits (2 different occasions), therefore the diagnosis of hypertension should not be made on the basis of a single BP measurement to begin with.

Response 1: Thank you for the comment. We fully agree with the Reviewer. While guidelines are crystal clear about this issue, the fact is that most (but not all) epidemiological studies (surveys) refer to “prevalence of hypertension” while BP was measured on only 1 visit (with subsequent substantial overestimation of the true prevalence of hypertension). The paper provides further evidence of the magnitude of the overestimation in papers on epidemiological survey that have assessed BP on only 1 visit. As we mention in the text, one can understand why many surveys only assess BP on 1 visit (for convenience reasons due to limited resources) and we recommend that such surveys should, at least, provide a clear discussion of this limitation, including by mentioning the range of the overestimation that this can incur (possibly also including that the title of such survey should include the “prevalence of high blood pressure” rather than “prevalence of hypertension”. We have also reworded a sentence in the introduction to make this point more explicit (on page 1 lines 38 – 39).

Point 2. In table 3, when divided into groups, the 95% confidence interval for the absolute difference in BP between visit 1 and visit 2 crosses 0, suggesting no statistical significance.  In general, statistical significance (or lack thereof) needs to be better indicated throughout the manuscript.

Response 2: Thank you for the comment. We agree that the intersection of 95%CIs of 2 estimates does not exactly correspond to statistical significance based on p-values, but several prominent epidemiologists advise to use 95CIs (vs p values) (e.g. Rothman, Morgenstern, Greenland). Of note, also, the issue in this study is not to assess if these differences are “significant” (which they largely are when based on overall estimates based on all participants), but rather to provide an order of magnitude of the relative difference between point estimates (i.e. “by how much the prevalence of HBP (based on readings on 1 visit) differs from the prevalence based on readings on 2 visits”.

Point 3. For table 4, the assumption the authors make about BP reduction in untreated individuals with elevated BP at visit 1 who did not have a second visit is unclear and I don’t think it’s valid (page 3 line 145).  Rather, the data should only be presented for those 50 individuals who had 2 visits.

Response 3: Thank you for the remark. The table provide the difference in the persons that had 2 visits. The model just implies that the difference found in these people would apply to the persons who did not attend a second visit. This assumption is used in analyses with missing data (imputation) and it is valid (given that it is only an assumption). Of note, this analysis aims at providing absolute numbers of persons that are wrongly labelled as hypertension (vs HBP) in a population (the magnitude of it in terms of public health impact), and not for deriving “statistical significance” values.

Reviewer 3 Report

The authors conducted an assessment of the prevalence in hypertension in a population based study and how 2 vs 1 visits can impact observations related to BP. The manuscript accomplished this using two distinct approaches; 1) a community based-screening program and 2) a narrative review of published manuscripts that also conducted community surveys.

Overall the manuscript is exceptionally well organized and written with clear tables that illustrate the various measures and present the data well. The conclusions from the community based-screening program are reasonable and the narrative review provides a unique layer and point of reflection.

The study really shines in its discussion as several paragraphs emphasize the various issues and pitfalls of the study and ways in which the study can benefit from including mention of the “white coat effect” and how it can influence BP readings. The manuscript requires little to no changes and it has clearly received a great deal of attention.

Author Response

Response: We thank the Reviewer for the positive comments.

Round 2

Reviewer 1 Report

The authors have addressed all of my major concerns.

Reviewer 2 Report

The authors addressed my comments in their reply.  I have no further comments.